# Synthesis, Biological Activity and Molecular Docking Studies of Novel Nicotinic Acid Derivatives

**DOI:** 10.3390/ijms23052823

**Published:** 2022-03-04

**Authors:** Kinga Paruch, Anna Biernasiuk, Dmytro Khylyuk, Roman Paduch, Monika Wujec, Łukasz Popiołek

**Affiliations:** 1Chair and Department of Organic Chemistry, Faculty of Pharmacy, Medical University of Lublin, 4A Chodźki Street, 20-093 Lublin, Poland; dmytro.khylyuk@umlub.pl (D.K.); monika.wujec@umlub.pl (M.W.); lukasz.popiolek@umlub.pl (Ł.P.); 2Chair and Department of Pharmaceutical Microbiology, Faculty of Pharmacy, Medical University of Lublin, 1 Chodźki Street, 20-093 Lublin, Poland; anna.biernasiuk@umlub.pl; 3Department of Virology and Immunology, Institute of Biological Sciences, Faculty of Biology and Biotechnology, Maria Curie-Skłodowska University, 19 Akademicka Street, 20-033 Lublin, Poland; rpaduch@poczta.umcs.lublin.pl

**Keywords:** *N*-acetyl-1,3,4-oxadiazoline derivatives, antimicrobial activity, cytotoxicity, molecular modelling, acylhydrazones

## Abstract

In our research, we used nicotinic acid as a starting compound, which was subjected to a series of condensation reactions with appropriate aldehydes. As a result of these reactions, we were able to obtain a series of twelve acylhydrazones, two of which showed promising activity against Gram-positive bacteria (MIC = 1.95–15.62 µg/mL), especially against *Staphylococcus epidermidis* ATCC 12228 (MIC = 1.95 µg/mL). Moreover, the activity of compound **13** against the *Staphylococcus aureus* ATCC 43300 strain, i.e., the MRSA strain, was MIC = 7.81 µg/mL. Then, we subjected the entire series of acylhydrazones to a cyclization reaction in the acetic anhydride, thanks to which we were able to obtain twelve new 3-acetyl-2,5-disubstituted-1,3,4-oxadiazoline derivatives. Obtained 1,3,4-oxadiazolines were also tested for antimicrobial activity. The results showed high activity of compound **25** with a 5-nitrofuran substituent, which was active against all tested strains. The most promising activity of this compound was found against Gram-positive bacteria, in particular against *Bacillus subtilis* ATCC 6633 and *Staphylococcus aureus* ATCC 6538 (MIC = 7.81 µg/mL) and ATCC 43300 MRSA strains (MIC = 15.62 µg/mL). Importantly, the best performing compounds did not show cytotoxicity against normal cell lines. It seems practical to use some of these compounds or their derivatives in the future in the prevention and treatment of infections caused by some pathogenic or opportunistic microorganisms.

## 1. Introduction

Antibiotic resistance describes the ability of bacteria to survive when exposed to antibiotics [1,2]. Once bacteria are exposed to antibiotics, there are three possibilities—the bacteria will die, stagnate (will not multiply) or will multiply. The third possibility is called antibiotic resistance, which poses a serious danger to public health worldwide. It leads to higher health care costs, longer hospitalization, patient failures and even deaths [3]. It turned out that it was necessary to introduce information and education activities in relation to medical professionals and to the entire society, as well as rigorous compliance with the rules of infection control and prevention of infections in all health care facilities (hospitals, outpatient facilities and nursing homes) [4]. Unfortunately, these actions are not sufficient and scientists are still constantly searching for new chemotherapeutic agents to help the immune system, while bacteria are constantly developing mechanisms that allow them to survive [5]. Bacterial strains resistant to all available antibiotics are already identified. Resistance to antibiotics is increasing faster than the pace of development of new pharmaceutics. Resistant microorganisms that pose a serious clinical and therapeutic problem include bacteria from *Enterobacteriaceae* family (*Klebsiella pneumoniae* and *Escherichia coli*) resistant to fluoroquinolones, third-generation cephalosporins, aminoglycosides and carbapenems, non-fermenting bacilli (*Acinetobacter* spp. and *Pseudomonas*), resistant to aminoglycosides and carbapenems, *Enterococcus* spp. resistant to vancomycin and linezolid and *Streptococcus pneumoniae* strain resistant to penicillin and the third-generation of cephalosporins. *Staphylococcus aureus* resistant to methicillin (MRSA), vancomycin (VISA) or linezolid is still a serious problem [6,7,8,9,10,11]. Particular attention is paid to golden staphylococcus (sometimes abbreviated as MRSA—methicillin-resistant *Staphylococcus aureus*) because it is a type of bacteria that more or less every third person transfers on the surface of the skin or in the nose, without becoming infected. However, *S. aureus* strain gets inside when skin is cut and can cause infection [12,13]. An antibiotic called methicillin is used to treat *S. aureus* infections. *Staphylococcus aureus* constitutes a variety of SA bacteria that are resistant to methicillin (and usually to some of the other antibiotics normally used to treat infections) [14,15,16]. Some microorganisms have acquired resistance to nearly all antibiotics. Therefore, the World Health Organization has published a list of antibiotic-resistant priority pathogens to which new antimicrobial agents are urgently needed. Some of them are listed in Table 1.

*Candida albicans* is one of the species of microorganisms that belong to the natural physiological flora of the human body. This fungus is isolated from the mucous membranes of the digestive tract, respiratory tract, mouth and skin. Unfortunately, these species can also cause opportunistic infections. First of all, these are non-dangerous, but troublesome to heal, surface yeasts [31]. Systemic candidiasis appears more and more often and constitutes a bigger problem. They are observed in patients with severe immunodeficiency, caused by long-term therapy or diseases which cause a decrease of the immune response, such as cancer or AIDS [32]. Unfortunately, in recent years, an increase in the number of isolated strains resistant to the antifungal agents used so far has been additionally observed. Moreover, some of them show resistance to many pharmaceutics at once, this is called multidrug resistance (MDR). This makes the candidiasis treatments ineffective [33].

Invasive *Candida* infections remain an important cause of morbidity and mortality, especially in hospitalized and immunocompromised or critically ill patients. Currently, there are only three major classes of medicines approved for the treatment of serious candidiasis. Moreover, the efficacy of these antimycotics is also compromised by the development of drug resistance in this pathogen population [29,30].

That is why it is so important to search for new molecules that will be able to fight even with the most resistant strains of bacteria or fungi and at the same time will be safe for the human body.

An important group of compounds that can constitute a good starting point for the development of new antibacterial and antifungal agents are acylhydrazones as well as 1,3,4-oxadiazole derivatives. Acylhydrazones have a wide spectrum of biological activity, including: antibacterial [34,35,36], antifungal [37], antitubercular [38] and anticancer properties [39,40]. Similarly, the derivatives of 1,3,4-oxadiazole have documented antibacterial [41,42,43,44], antifungal [45,46], anticancer [47,48] and antimycobacterial [49,50] action. Therefore, our research goal was based on the synthesis of appropriate acylhydrazones which subsequently were transformed into 3-acetyl-2,5-disubstituted-1,3,4-oxadiazole derivatives. The starting compound for our syntheses was nicotinic acid hydrazide. Nicotinic acid is known collectively as vitamin B3 or vitamin PP. It has no proven antibacterial effect, but vitamin B3 increases the ability of immune cells to fight against *Staphylococcus aureus* bacteria because it increases the number of white blood cells (neutrophils), which possess a bactericidal effect [51].

Having in mind these facts, in this research we decided to design, synthetize, evaluate for antimicrobial activity and cytotoxicity and perform molecular docking studies of acylhydrazones and 1,3,4-oxadiazole derivatives obtained from nicotinic acid hydrazide.

## 2. Results

### 2.1. Chemistry

In this paper, we present the synthesis of a series of new acylhydrazones (**2**–**13**) and 3-acetyl-2,5-disubstituted-1,3,4-oxadiazoline derivatives (**14**–**25**) (Figure 1). Novel molecules were obtained with a 63–96% yield. The 1,3,4-oxadiazoline derivatives were obtained with a lower yield. The presented synthesis was carried out in two stages. Initially, a series of new acylhydrazones (**2**–**13**) was obtained as a result of the condensation reaction of nicotinic acid hydrazide (**1**) with the appropriate aldehydes. Then, the obtained compounds (**2**–**13**) were subjected to the cyclization reaction in acetic anhydride in order to synthesize novel 1,3,4-oxadiazoline derivatives. All synthesized compounds are solids and can be dissolved in DMSO at room temperature.

The structure of all obtained compounds was confirmed by spectroscopic methods: ^1^H NMR, ^13^C NMR and FT-IR spectra as well as elemental analysis.

The first group of compounds, i.e., acylhydrazones (**2**–**13**), showed the following characteristic signals in the ^1^H NMR spectra: singlet for the NH group in the range of 11.73–12.91 ppm and the signal for the =CH group at δ 8.23–9.11 ppm. In the case of the ^13^C NMR spectra, the signals for the carbon atom of =CH and C=O groups appeared in the range of δ 137.11–138.61 ppm and 161.18–185.41 ppm, respectively. Additionally, we observed characteristic signals in expected regions in the FT-IR spectra. The remaining fragments of the examined compounds gave characteristic signals in the expected ranges of the chemical shift.

The 3-acetyl-2,5-disubstituted-1,3,4-oxadiazoline derivatives (**14**–**25**) in the ^1^H NMR, showed a singlet signal for the proton of CH group of the 1,3,4-oxadiazoline system at δ 7.15–10.01 ppm and for the protons of the acetyl substituent in the range of δ 1.93–2.57 ppm. Signals for the carbon atom of the CH group of 1,3,4-oxadiazoline system and for the carbon atom of 1,3,4-oxadiazoline ring in the ^13^C NMR spectra were found at δ 88.41–125.08 ppm and δ 150.83–155.08 ppm, respectively. Similarly, the signal for the carbon atom of the methyl group of the acetyl substituent appeared around 20 ppm. Additionally, we observed characteristic signals for C=O, C=N and C-OC bonds at the expected values in FT-IR spectra.

### 2.2. Microbiology

All synthesized compounds (**2**–**25**) were examined for their antimicrobial activity against Gram-positive bacteria, Gram-negative bacteria and fungi belonging to yeasts *Candida* spp. A panel of reference strains of microorganisms also included some resistant staphylococci, that is the methicillin-resistant *Staphylococcus aureus* MRSA ATCC 43300. On the basis of MIC and MBC values, we discovered that new compounds showed interesting antibacterial activity. Table 2 presents the compounds which displayed antimicrobial activity.

The obtained results showed a very high antibacterial effect of compounds **5** and **13** against Gram-positive bacteria (Table 2). Their activity was strong or very strong with minimal inhibitory concentrations (MIC) ranging from 7.81 µg/mL to 15.62 µg/mL and from 1.95 µg/mL to 15.62 µg/mL, respectively. In turn, ranges of minimal bactericidal concentrations (MBC) values were 7.81–31.25 µg/mL for **5** and 3.91–31.25 µg/mL for compound **13**. Moreover, these substances showed a bactericidal effect (MBC/MIC = 1–4).

Additionally, two of the tested compounds showed activity against the MRSA strain *Staphylococcus aureus* ATCC 43300 (compound **5**: MIC = 15.62 µg/mL, compound **13**: MIC = 7.81 µg/mL).

Gram-negative bacteria were less sensitive to these compounds. Acylhydrazone numbered as **13** had good or moderate bactericidal activity against all rod-shaped bacteria with MIC = 31.25–500 µg/mL and MBC = 31.25–1000 µg/mL (MBC/MIC = 1–4). In turn, compound **5** indicated activity towards *Bordetella bronchiseptica* ATCC 4617 (MIC = 62.5 µg/mL, MBC = 125 µg/mL and MBC/MIC = 2). The remaining bacteria were insensitive to these compounds. Other newly synthesized compounds showed some activity against staphylococci or micrococci (only compounds **7** and **9**) or were inactive towards reference bacteria.

The yeasts, belonging to reference *Candida* spp. were sensitive only to compound **5**. Antifungal activity of **5** was good or moderate (MIC = 62.5–250 µg/mL and MFC = 500–1000 µg/mL) with fungicidal (MFC/MIC = 4) or fungistatic (MFC/MIC = 8) effect.

In the case of the 3-acetyl-2,5-disubstituted-1,3,4-oxadiazoline derivatives, the widest spectrum of activity, with bactericidal and fungicidal effect (MBC/MIC = MFC/MIC = 1–2), was shown by derivative **25** towards all reference bacteria and yeasts (Table 2). Gram-positive bacteria were the most sensitive (MIC = 7.81–62.5 µg/mL and MBC = 7.81–125 µg/mL). Compounds numbered as **15**, **17**, **25** were active against the MRSA strain *Staphylococcus aureus* ATCC 43300. Their activity was strong to moderate (MIC = 15.62–250 µg/mL). Compound **25** indicated good effect towards Gram-negative microorganisms with MIC = 31.25–125 µg/mL and MBC = 62.5–125 µg/mL towards all strains except *Pseudomonas aeruginosa* ATCC 9027 (moderate activity; MIC = MBC = 250 µg/mL). Gram-negative bacteria were sensitive only to this compound.

Moreover, the 1,3,4-oxadiazoline derivative **17** had bactericidal effect (MBC/MIC = 1–4) with very strong or strong activity (MIC = 7.81–15.62 µg/mL, MBC = 7.81–62.5 µg/mL) against *Staphylococcus epidermidis* ATCC 12228 or *Staphylococcus aureus* ATCC 6538 and good activity towards other Gram-positive bacteria (MIC = 31.25–62.5 µg/mL, MBC = 31.25–125 µg/mL). The remaining newly synthesized compounds (**14**, **15**, **16**, **23** and **24**) were less active against cocci and bacilli or were inactive towards reference strains of bacteria.

Compound **25** also indicated a high fungicidal effect towards *Candida* spp. with MIC values ranging from 15.62 µg/mL to 500 µg/mL and MFC from 31.25 µg/mL to 500 µg/mL). Yeasts belonging to *C. albicans* ATCC 10231 and *C. parapsilosis* ATCC 22019 were especially sensitive to this substance (MIC = 15.62 µg/mL and two-fold higher MFC value). The molecule **24** was slightly less active (moderate activity; MIC = 500 µg/mL and MFC = 500–1000 µg/mL). The antifungal effect of some other substances was moderate or mild (**14**, **15**, **18** and **23**). The remaining compounds were inactive towards fungi.

### 2.3. Cytotoxicity Studies

The analysis of cells’ sensitivity to the tested compounds was performed with the use of the MTT method—analysis of the metabolic activity of cells and NR uptake assay analysing the stability of cell membranes and thus direct cytotoxicity.

The MTT method showed a higher sensitivity of colon epithelial tumour cells to the analysed compounds than normal cells of the colonic epithelium. Compound **21** had the weakest effect on tumour cells, reducing their metabolic activity to 80% at the compound concentration of 200 µg/mL. The strongest activity was shown by compound **17**, which reduced the metabolic activity of cells to 41% at the concentration of 75 µg/mL in comparison to the control (100% activity), not treated with the tested substances. The IC_50_ value, in this case, was 61.18 µg/mL. In the case of normal cells, the highest decrease in cell metabolic activity was also demonstrated after compound **17** application. However, at the concentration of 200 µg/mL, cell metabolism decreased only to 71.6% as compared with the untreated control (Figure 1).

Carrying out the analysis by the neutral red uptake (NR), there were slight differences in the sensitivity of the cells to the tested compounds in comparison to the results obtained with the MTT method. In the case of tumour cells, all tested compounds showed similar cytotoxic activity. At the highest concentration of the compounds (200 µg/mL), HT29 cell viability oscillated between 70% and 78%. In the case of normal colonic epithelial cells, the compound numbered **20** maintained cell viability over the control value at the entire range of tested concentrations. Compounds **21** and **22** at the concentration of 200 µg/mL decreased cell viability to 85 and 80%, respectively. Compound **17** showed the strongest cytotoxic activity against these cells, reducing their viability to 46% at the highest applied concentration (200 µg/mL). This allowed the calculation of an IC_50_ value for this compound against normal cells at 179.81 µg/mL (Figure 2).

The tested compounds (**17**, **20**, **21**, **22**) stimulated the production of nitric oxide (NOx) by both normal and tumour cells derived from the human colonic epithelium. However, a different reaction of both types of cells to increasing concentrations of the tested compounds can be observed. In the case of analyses performed on cancer cells, the lowest concentrations of the tested compounds (25 µg/mL), significantly stimulated the production and release of nitric oxide to values exceeding 0.5 µM as compared to the control (0.147 µM). The increasing concentrations of the tested compounds (**17**, **20**, **21**, **22**) were less effective in inducing the release and production of nitric oxide. The highest concentration (200 µg/mL) of the compounds led to the production of nitric oxide by tumour cells at the level of 0.372 µM (**17**) to 0.101 µM (**22**). Analysing normal cells, it was shown that the tested substances stimulated the release of nitric oxide with increasing concentration of the compound. Substance **17** had the strongest stimulating effect on the production of NOx by normal cells. At the highest concentration (200 µg/mL) it stimulated the release of NOx to a value of 0.566 µM, while the weakest action was found for compound **22**, which at the highest concentration stimulated the release of NOx to a level of 0.419 µM (Figure 3).

The reduction abilities of the tested compounds (**17**, **20**, **21**, **22**) were assessed by DPPH and FRAP methods. The results of the obtained studies were compared to known compounds with reductive activity, water-soluble synthetic vitamin E (Trolox) and ascorbic acid, respectively. In general, the tested compounds (**17**, **20**, **21**, **22**) showed weak but detectable reductive activity. Compound **17** was the most effective, reducing the DPPH free radical at a concentration of 150 µg/mL which is equivalent to that of synthetic vitamin E at a concentration of 3.558 ± 0.717 µg/mL. The weakest reducing activity was observed for compound **20**. On the other hand, compound **20** showed the strongest ferric-reducing power (FRAP), equal to 3.51 ± 1.35 µg/mL of ascorbic acid activity. The weaker reducing activity of the Fe^3+^ ion was found for compound **22** (Table 3 and Table 4).

May–Grünwald–Giemsa (MGG) cell staining was designed to evaluate cell morphology after incubation for 24 h with compounds **17**, **20**, **21** and **22**. The performed observations are in line with the quantitative results obtained with the NR method. The activity of compounds determined to be cytotoxic in NR uptake assay, in MGG staining in both normal and neoplastic cells was observed as a decrease in the number of stained cells due to the detachment of dying cells from the carrier surface. In addition, you can also observe the detachment of cells from each other, their shrinkage as well as loss of intercellular interaction (Appendix A).

### 2.4. Molecular Docking

In order to predict the target enzymes for most active compounds (**5**, **13**, **17**, **25**) and to evaluate the possible interactions, a molecular docking study was carried out with the use of the Autodock Vina 4.2 [52]. Structures of compounds **13** and **25** are chemically related to widespread chemotherapeutic agents, which contain 5-nitrofuran moieties, such as Furazolidone, Nitrofurazone, and Nifuroxazide [53]. That is why it is logical to suppose that synthesized compounds may have the same mode of antibacterial action. This purpose is also confirmed by the fact that the structurally related compounds with 5-nitrophenyl moiety **10** and **22** do not possess any antimicrobial activity. The 5-nitrofuran derivatives have to be activated before mediating its cytotoxic effects. Reactions of activations are catalyzed by nitroreductase (NTR) enzymes. As a result, the production of free radicals is observed. They can readily react with cellular macromolecules, and they are directly responsible for antibacterial action. As a result, lipids oxidation, cell membrane damages, enzyme inactivation, and finally fragmentation of the DNA sequence is observed [54]. That is why we evaluated the binding energy of compounds **5** and **17** for *Escherichia coli* Nitroreductase (PDB code: 1YKI) and compare them with such energy for native ligand Nitrofurazone. Additionally, we made a cross-docking simulation with the nitrofurantoin, which was used as the reference compound in the microbiology assays. The dihydrofolate reductase (*S. aureus*, PDB code: 5ISP) and tyrosyl-tRNA synthetase (*S. aureus*, PDB code: 1JIJ) were also selected for *in silico* docking simulations of antibacterial activity of the most active compounds (**5**, **13**, **17**, **25**).

Docking demonstrated that compounds **13** and **25** have a better affinity to Nitroreductase compared to Nitrofurazone (Table 5).

As shown in Figure 4, compound **13** linked tightly with amino acid residues of *Escherichia coli* Nitroreductase by hydrogen bonds with LYS74 (2.54Å), LYS14 (2.42 Å), GLU165 (2.11Å), THR41 (2.15Å), ARG121 (2.57Å) and ASN117 (2.30Å). Binding energy also increases owing to additional stabilized forces, such as van der Waals, attractive charge and amide-Pi stacked interactions between molecule and ASN42, GLU165 and THR41, respectively.

Compound **25** formed a number of hydrogen bonds with GLU162 (2.52Å), ASN71 (2.33Å), GLY166 (1.91Å), LYS14 (2.50Å) and ARG107 (2.63Å). Additionally, there are different forms of hydrophobic interaction between PHE 124, ASN42 and GLU102.

It is quite similar to the interaction between Nitrofurazone and the Nitroreductase active site, where planar medicine molecule also makes several hydrogen bonds with the protein [55]. Docking results allowed us to suppose that compounds **13** and **25** would be more attractive substrates for Nitroreductase of *S. aureus* and interesting scaffold for new 5-nitrofuran derivatives with a wide spectrum of antimicrobial activities (Figure 5).

Compounds **5** and **17** have quite a good affinity to target enzymes, nevertheless, those energies are smaller when compared to binding energies of initial ligands. Docking results allow suggesting that antimicrobial activity of substances **5** and **17** is connected mainly with inhibition of tyrosyl-tRNA synthetase. The most active compounds (**13** and **25**) showed good affinity to tyrosyl-tRNA synthetase and it is correlated with their wide activity against either Gram-positive or Gram-negative bacteria strains. The binding energies are shown in Table 6.

The planar molecule of compound **5** occupies hydrophobic pockets, made by LEU8, VAL31 and LEU54. Additionally, three amino acids have alkyl interactions with iodine substituent of the phenyl ring (TYR98, ILE14 and VAL6) (Figure 6).

Compound **17** makes the several alkyl interactions with aliphatic aminoacids LEU20, ALA7, VAL20 and VAL31, also by the iodine atoms, but the lack of hydrogen bond interaction decreases the affinity of compounds **5** and **17** with tyrosyl-tRNA synthetase, compared to the reference ligand SB-239629 (Figure 7) [56].

An appropriate ADMET profile for new compounds is often crucial for further development as new potential antimicrobial agents. Modern in silico techniques allow simplifying the search for new chemical entities with suitable physicochemical properties for efficient and safe oral administration. The Lipinski rule of five become the non-official standard in medicinal chemistry. Therefore, ADMET profile predictions were made for **5**, **13**, **17** and **25** using freely accessible portals SwissADME and ProTOX. Additionally, ciprofloxacin and nitrofurantoin were chosen as the reference compounds for comparison. Results are highlighted in Table 7.

According to obtained data, compounds **5**, **13**, **17** and **25** possess suitable ADMET profiles, which allows us to suggest them as the perspective antimicrobial agents.

## 3. Discussion

The condensation reaction is a simple method by which acylhydrazones (**2**–**13**) can be obtained quickly. Then, the cyclization reaction in acetic anhydride of obtained compounds enabled the synthesis of 3-acetyl-2,5-disubstituted-1,3,4-oxadiazoline derivatives. The ease of this synthesis, good yield results and high biological potential of the obtained derivatives encourage the search for compounds of antimicrobial nature among them. Among the compounds from the group of acylhydrazones, a substance with the 5-nitrofuran substituent showed the highest activity. It showed high activity especially against Gram-positive bacterial strains, among them towards *Staphylococcus epidermidis* (MIC = 1.95 µg/mL), *Staphylococcus aureus* ATCC 6538 (MIC = 3.91 µg/mL) and also against the MRSA strain (MIC = 7.81 µg/mL). The 1,3,4-oxadiazoline derivative, i.e., the compound numbered as **25**, showed half of the activity (MIC = 15.62 µg/mL) against the same strains. Similarly, compound **5** with the 2-hydroxy-3,5-diiodophenyl substituent after conversion to 1,3,4-oxadiazoline derivative showed half of the activity against the same bacterial strains. On the basis of obtained results, it can be concluded that the nicotinic acid-derived acylhydrazones are more active than the corresponding 1,3,4-oxadiazoline derivatives in terms of antibacterial activity against Gram-positive bacteria. On the other hand, the activity against fungal strains was different. In this case, the 3-acetyl-1,3,4-oxadiazolines were more active towards yeasts than acylhydrazones. This fact can be observed in the example of the compounds with the 5-nitrofuran substituent, i.e., **13** (acylhydrazone) and **25** (1,3,4-oxadiazoline derivative). The substance **25** showed good activity against *Candida albicans* ATCC 10231 (MIC = 15.62 µg/mL), while compound **13** showed no activity.

Analysis of the results of the conducted research in terms of the structure–activity relationship (SAR), it can be stated that acylhydrazones showed greater biological activity in relation to *N*-acetyl-1,3,4-oxadiazolines. The most active of the two groups were those derivatives that had the 5-nitrofuran substituent. We saw a similar situation in our previous articles [57–61, where, the cyclization process also resulted in a decrease in the activity of the tested compounds. Thus, it can be concluded that the introduction of a 1,3,4-oxadiazole ring into the molecule deteriorates the antimicrobial activity, and the 5-nitro-furoyl moiety significantly improves the effectiveness against bacteria and fungi.

Moreover, the tested acylhydrazones were significantly more active against Gram-positive bacteria than against Gram-negative bacteria. This difference can be seen perfectly during the analysis of the activity of compound **5** with a 2-hydroxy-3,5-diiodophenyl substituent, where the activity against Gram-positive bacteria was within the limits of MIC = 7.81–15.62 µg/mL and the activity against Gram-negative was observed only for one strain—*Bordetella bronchiseptica* ATCC 4617 (MIC= 62.5 µg/mL), and the others substances were not active. A similar difference in activity was observed for the compound numbered **13** with a 5-nitrofuran substituent. It was the most active against Gram-positive bacteria *Staphylococcus epidermidis* ATCC 12228 (MIC = 1.95 µg/mL), and against the remaining Gram-positive, it also showed high activity (MIC =1.95–15.62 µg/mL). On the other hand, against Gram-negative strains, activity was either good or moderate. The activity of *N*-acetyl-1,3,4-oxadiazolines was similar to acylhydrazones. They also showed greater activity against Gram-positive than towards Gram-negative bacteria. The most active of this group was compound numbered as **25** against Gram-positive strains at (MIC = 7.81–62.5 µg/mL), while for Gram-negative bacteria the range of activity was MIC = 62.5–250 µg/mL. Additionally, for compound **17**, the measured activity against Gram-positive bacteria was MIC = 7.81–62.5 µg/mL, and no activity was found towards Gram-negative strains.

In conclusion, the results showed slightly different activity of the newly synthesized compounds against Gram-positive and Gram-negative bacteria. Among them, compounds **5**, **13**, **17** and **25** had a satisfactory and beneficial activity with strong or good effect towards all reference Gram-positive bacteria. In turn, microorganisms belong to bacteria from *Enterobacterales* and non-fermenting rods were less sensitive to these substances. Only substances **13** and **25** showed some activity with moderate or good effects depending on the microorganism. Due to the distinctive structure of the cell wall, Gram-negative bacteria are usually more resistant to antibiotics or other antibacterial interventions than Gram-positive. Their cell membrane is thin but difficult to penetrate. Therefore, these bacteria are harder to kill [57]. Our results also confirm this relationship.

## 4. Materials and Methods

### 4.1. Chemistry

All reagents used for the experiments were purchased from Merck Co. (Darmstadt, Germany) and used without further purification. They had a class of purity declared by the manufacturer. The melting points of the obtained compounds were determined with a Fisher–Johns apparatus (Fisher Scientific, Schwerte, Germany), and presented without any correction. The purity of the obtained compounds was assessed through thin layer chromatography (TLC) on plates covered with silica gel (aluminium oxide 60 F-254) by Merck. Chloroform–ethanol mixtures in the ratio 10:1 (*v/v*) were used as the mobile phase. Spots were developed by irradiation with UV light with a wavelength λ = 254 nm. The FT-IR spectra were recorded on a Nicolet 6700 spectrometer (Thermo Scientific, Madison, Wisconsin, USA) in cm^−1^. The ^1^H and ^13^C NMR spectra were recorded on the Bruker Avance 300 and 600 apparatus (Bruker BioSpin GmbH, Ettlingen, Germany). The compounds were dissolved in dimethyl sulfoxide (DMSO-*_d6_*) before analysis. Tetramethylsilane (TMS) was used as an internal standard. Chemical shift values are given in ppm. Elemental analysis was determined by a Perkin Elmer 2400 Series II CHNS/O analyzer (Waltham, MA, USA), and the results were within ±0.4% of the theoretical values.

#### 4.1.1. Synthesis of Acylhydrazones of Nicotinic Acid Hydrazide (**2**–**13**)

New acylhydrazones of nicotinic acid (**2**–**13**) were obtained on the basis of the procedure reported earlier [58,59,60,61,62]. The nicotinic acid hydrazide (**1**) (0.01 mole) was dissolved in 20 mL of ethanol (96%) and appropriate aldehyde (0.011 mole) was added. The mixture was heated under reflux for 3 h. Subsequently, the solution was allowed to cool and placed in the refrigerator for 24 h. The precipitate was filtered off and recrystallized from ethanol.

Physicochemical properties of acylhydrazones of nicotinic acid (**2**–**13**):

*N*-[(4-*tert*-butylphenyl)methylidene]pyridine-3-carbohydrazide (**2**)

White powder, yield: 87%, M.p.: 116 °C; IR: 3443, 3220 (N-H), 3031 (CH, arom.), 2958 (CH, aliph.), 1647 (C=O), 1563 (C=N), 1295, 1151 (C-OC), 1028 (N-N); ^1^H NMR (600 MHz, DMSO-*d_6_*) δ (ppm): 1.31 (s, 9H, 3xCH_3_), 7.50–7.51 (d, 2H, ArH, *J* = 6 Hz), 7.57–7.59 (m, 1H, ArH), 7.68–7.69 (d, 1H, ArH, *J* = 6 Hz), 8.25–8.27 (m, 1H, ArH), 8.43 (s, 1H, =CH), 8.77–8.78 (m, 1H, ArH), 9.06–9.07 (m, 1H, ArH), 11.98 (s, 1H, NH); ^13^C NMR (150 MHz, DMSO-*d_6_*) δ (ppm): 31.78 (3xCH_3_), 35.70 (C*_t_*_-butyl_), 124.06, 126.15, 127.51, 129.71, 131.88 (7C_ar_), 135.88 (=CH), 148.87, 149.02, 152.70, 153.57 (4C_ar_), 162.07 (C=O); Anal. calc. for C_17_H_19_ClN_3_O (281.35) (%): C 72.57; H 6.81; N 14.94. Found: C 77.25; H 6.31; N 14.50.

*N*-[(3-bromo-4-hydroxyphenyl)methylidene]pyridine-3-carbohydrazide (**3**)

White powder, yield: 93%, M.p.: 260 °C; IR: 3458 (OH), 3189 (N-H), 3021 (CH, arom.), 2970 (CH, aliph.), 1636 (C=O), 1597 (C=N), 1229, 1080 (C-OC), 1017 (N-N); ^1^H NMR (600 MHz, DMSO-*d_6_*) δ (ppm): 7.49–7.51 (m, 1H, ArH), 7.56–7.59 (m, 1H, ArH), 7.94 (s, 1H, OH), 8.15–8.16 (m, 1H, ArH), 8.24–8.26 (m, 1H, ArH), 8.30 (s, 1H, =CH), 8.69–8.70 (m, 1H, ArH), 8.77–8.78 (m, 1H, ArH), 9.06–9.07 (d, 1H, ArH, *J* = 6 Hz), 12.13 (s, 1H, NH); ^13^C NMR (150 MHz, DMSO-*d_6_*) δ (ppm): 112.69, 124.05, 129.14, 129.30, 129.53, 131.21, 135.13 (7C_ar_), 135.94 (=CH), 145.83, 149.05, 152.23, 152.78 (4C_ar_), 162.20 (C=O); Anal. calc. for C_13_H_10_BrN_3_O_2_ (320.14) (%): C 48.77; H 3.15; N 24.96. Found: C 49.25; H 3.31; N 25.50.

*N*-[(3,5-dibromo-4-hydroxyphenyl)methylidene]pyridine-3-carbohydrazide (**4**)

White powder, yield: 92%, M.p.: 242 °C; IR: 3528 (OH), 3294 (N-H), 3062 (CH, arom.), 2970 (CH, aliph.), 1651 (C=O), 1597 (C=N), 1293, 1067 (C-OC), 1029 (N-N); ^1^H NMR (600 MHz, DMSO-*d_6_*) δ (ppm): 7.03–7.04 (d, 1H, ArH, *J* = 6 Hz), 7.56–7.60 (m, 1H, ArH), 7.88–7.89 (d, 1H, ArH, *J* = 6 Hz), 8.24–8.26 (m, 1H, ArH), 8.31 (s, 1H, =CH), 8.76–8.77 (m, 1H, ArH), 9.05–9.06 (d, 1H, ArH, *J* = 6 Hz), 10.85 (s, 1H, OH), 11.97 (s, 1H, NH); ^13^C NMR (150 MHz, DMSO-*d_6_*) δ (ppm): 110.31, 117.03, 124.05, 127.30, 128.47, 129.69, 131.97 (7C_ar_), 135.87 (=CH), 147.53, 149.00, 152.68, 156.45 (4C_ar_), 162.00 (C=O); Anal. calc. for C13H9Br_2_N_3_O_2_ (399.04) (%): C 39.13; H 2.27; N 10.53. Found: C 39.27; H 2.31; N 10.83.

*N*-[(2-hydroxy-3,5-diiodophenyl)methylidene]pyridine-3-carbohydrazide (**5**)

Yellow powder, yield: 89%, M.p.: 222 °C; IR: 3426 (OH), 3221 (N-H), 3016 (CH, arom.), 2970 (CH, aliph.), 1652 (C=O), 1513 (C=N), 1272, 1045 (C-OC), 1022 (N-N); ^1^H NMR (600 MHz, DMSO-*d_6_*) δ (ppm): 7.60–7.62 (m, 1H, ArH), 7.93–7.94 (d, 1H, ArH, *J* = 6 Hz), 8.07–8.08 (d, 1H, ArH, *J* = 6 Hz), 8.29–8.30 (m, 1H, ArH), 8.45 (s, 1H, =CH), 8.80–8.81 (m, 1H, ArH), 9.10–9.11 (d, 1H, ArH, *J* = 6 Hz), 12.71 (s, 1H, OH), 12.82 (s, 1H, OH); ^13^C NMR (150 MHz, DMSO-*d_6_*) δ (ppm): 82.65, 88.29, 120.70, 124.16, 128.56 (5C_ar_), 136.08 (=CH), 139.36, 147.18, 148.20, 149.18, 153.22, 157.11 (6C_ar_), 162.17 (C=O); Anal. calc. for C_13_H_9_I_2_N_3_O_2_ (493.04) (%): C 31.67; H 1.84; N 8.52. Found: C 32.15; H 1.96; N 8.74.

*N*-[2-(2-bromo-3-hydroxy-4-methoxyphenyl)ethylidene]pyridine-3-carbohydrazide (**6**)

Grey powder, yield: 91%, M.p.: 220 °C; IR: 3522 (OH), 3163 (N-H), 3003 (CH, arom.), 2967 (CH, aliph.), 1641 (C=O), 1491 (C=N), 1251, 1063 (C-OC), 1028 (N-N); ^1^H NMR (600 MHz, DMSO-*d_6_*) δ (ppm): 3.90 (s, 3H, CH_3_), 7.11–7.12 (d, 1H, ArH, *J* = 6Hz), 7.53–7.54 (d, 1H, ArH, *J* = 6 Hz), 7.57–7.59 (m, 1H, ArH), 8.26–8.28 (m, 1H, ArH), 8.76–8.77 (d, 1H, ArH, *J* = 6 Hz), 8.78 (s, 1H, =CH), 9.08–9.09 (d, 1H, ArH, *J* = 6 Hz), 9.71 (s, 1H, OH), 12.12 (s, 1H, NH); ^13^C NMR (150 MHz, DMSO-*d_6_*) δ (ppm): 111.50, 112.52, 118.25, 124.03, 126.17, 129.60 (6C_ar_), 135.89 (=CH), 144.24, 147.99, 149.06, 150.20, 152.73 (5C_ar_), 162.01 (C=O); Anal. calc. for C_15_H_14_BrN_3_O_3_ (364.19 (%): C 49.47; H 3.87; N 11.54. Found: C 49.25; H 3.51; N 11.50.

*N*-[(2-bromo-5-fluorophenyl)methylidene]pyridine-3-carbohydrazide (**7**)

White powder, yield: 95%, M.p.: 128 °C; IR: 3442, 3190 (N-H), 3056 (CH, arom.), 2970 (CH, aliph.), 1656 (C=O), 1570 (C=N), 1297, 1103 (C-OC), 1028 (N-N); ^1^H NMR (600 MHz, DMSO-*d_6_*) δ (ppm): 7.30–7.33 (m, 1H, ArH), 7.59–7.61 (m, 1H, ArH), 7.71–7.73 (m, 1H, ArH), 7.76–7.79 (m, 1H, ArH), 8.28–8.30 (m, 1H, ArH), 8.78 (s, 1H, =CH), 8.79–8.80 (m, 1H, ArH), 9.10–9.11 (m, 1H, ArH), 12.38 (s, 1H, NH); ^13^C NMR (150 MHz, DMSO-*d_6_*) δ (ppm): 113.62, 113.78, 118.66, 119.70, 124.10, 129.25, 135.56, 135.61 (8C_ar_), 135.99 (=CH), 146.04, 149.12, 153.00 (3C_ar_), 162.34 (C=O); Anal. calc. for C_13_H_9_BrFN_3_O (322.13) (%): C 48.47; H 2.82; N 13.04. Found: C 49.27; H 2.92; N 13.50.

*N*-[(4-bromo-2-fluorophenyl)methylidene]pyridine-3-carbohydrazide (**8**)

Grey powder, yield: 85%, M.p.: 168 °C; IR: 3374, 3179 (N-H), 3047 (CH, arom.), 2967(CH, aliph.), 1647 (C=O), 1556 (C=N), 1290, 1068 (C-OC), 1025 (N-N); ^1^H NMR (600 MHz, DMSO-*d_6_*) δ (ppm): 7.54–7.56 (m, 1H, ArH), 7.58–7.60 (m, 1H, ArH), 7.70–7.72 (m, 1H, ArH), 7.89–7.91 (t, 1H, ArH, *J* = 6 Hz), 8.26–8.28 (m, 1H, ArH), 8.63 (s, 1H, =CH), 8.78–8.79 (m, 1H, ArH), 9.08–9.09 (d, 1H, ArH, *J* = 6 Hz), 12.20 (s, 1H, ArH); ^13^C NMR (150 MHz, DMSO-*d_6_*) δ (ppm): 119.88, 120.04, 121.67, 124.13, 128.26, 128.85, 129.35 (7C_ar_), 135.93 (=CH), 140.69, 149.06, 152.94, 160.09 (4C_ar_), 162.18 (C=O); Anal. calc. for C_13_H_9_BrFN_3_O (322.13) (%): C 48.47; H 2.82; N 13.04. Found: C 48.99; H 3.13; N 12.99.

*N*-[(2-chloro-6-fluorophenyl)methylidene]pyridine-3-carbohydrazide (**9**)

White powder, yield: 88%, M.p.: 186 °C; IR: 3462,3183 (N-H), 3028 (CH, arom.), 2970 (CH, aliph.), 1739 (C=O), 1601 (C=N), 1227, 1070 (C-OC), 1036 (N-N); ^1^H NMR (600 MHz, DMSO-*d_6_*) δ (ppm): 7.35–7.38 (m, 1H, ArH), 7.44–7.46 (d, 1H, ArH, *J* = 12 Hz), 7.49–7.53 (m, 1H, ArH), 7.58–7.60 (m, 1H, ArH), 8.28–8.29 (d, 1H, ArH, *J* = 6 Hz), 8.71 (s, 1H, =CH), 8.79–8.80 (d, 1H, ArH, *J* = 6 Hz), 9.09–9.10 (d, 1H, ArH, *J* = 6 Hz), 12.24 (s, 1H, NH); ^13^C NMR (150 MHz, DMSO-*d_6_*) δ (ppm): 116.05, 116.20, 124.09, 126.58, 129.37, 132.34, 134.41 (7C_ar_), 135.99 (=CH), 142.28, 149.12, 152.92, 159.82 (4C_ar_), 162.21 (C=O); Anal. calc. for C_13_H_9_ClFN_3_O (277.68) (%): C 56.23; H 3.27; N 15.13. Found: C 55.89; H 3.31; N 15.23.

*N*-[(2-chloro-5-nitrophenyl)methylidene]pyridine-3-carbohydrazide (**10**)

White powder, yield: 87%, M.p.: 220 °C; IR: 3296 (N-H), 3098 (CH, arom.), 2927 (CH, aliph.), 1663 (C=O), 1508 (C=N), 1298, 1042 (C-OC), 1022 (N-N); ^1^H NMR (600 MHz, DMSO-*d_6_*) δ (ppm): 7.60–7.62 (m, 1H, ArH), 7.86–7.88 (d, 1H, ArH, *J* = 12 Hz), 8.26–8.28 (m, 1H, ArH), 8.30–8.31 (m, 1H, ArH), 8.73–8.74 (d, 1H, ArH, *J* = 6 Hz), 8.80–8.81 (m, 1H, ArH), 8.88 (s, 1H, =CH), 9.11–9.12 (d, 1H, ArH, *J* = 6 Hz), 12.45 (s, 1H, NH); ^13^C NMR (150 MHz, DMSO-*d_6_*) δ (ppm): 121.66, 124.16, 125.88, 129.12, 132.11, 133.37 (6C_ar_), 136.02 (=CH), 139.66, 142.80, 147.21, 149.12, 153.11 (5C_ar_), 162.43 (C=O); Anal. calc. for C_13_H_9_ClN_4_O_3_ (304.69) (%): C 51.25; H 2.98; N 18.39. Found: C 49.88; H 2.51; N 18.99.

*N*-[(2,3-difluorophenyl)methylidene]pyridine-3-carbohydrazide (**11**)

White powder, yield: 96%, M.p.: 146 °C; IR: 3385, 3167, (N-H), 3069 (CH, arom.), 2990 (CH, aliph.), 1664 (C=O), 1561 (C=N), 1299, 1081 (C-OC), 1029 (N-N); ^1^H NMR (600 MHz, DMSO-*d_6_*) δ (ppm): 7.31–7.34 (m, 1H, ArH), 7.52–7.56 (m, 1H, ArH), 7.59–7.61 (m, 1H, ArH), 7.76–7.77 (m, 1H, ArH), 8.27–8.29 (d, 1H, ArH, *J* = 12 Hz), 8.68 (s, 1H, =CH), 8.79–8.80 (m, 1H, ArH), 9.09–9.10 (d, 1H, ArH, *J* = 6 Hz), 12,24 (s, 1H, NH); ^13^C NMR (150 MHz, DMSO-*d_6_*) δ (ppm): 119.15, 119.26, 122.11, 124.13, 124.50, 125.73, 129.31 (7C_ar_), 135.95 (=CH), 140.61, 149.08, 151.25, 152.97 (4C_ar_), 162.24 (C=O); Anal. calc. for C_13_H_9_FN_3_O (261.23) (%): C 59.77; H 3.47; N 16.09. Found: C 6.01; H 3.52; N 15.89.

*N*-{[4-(pyrrolidin-1-yl)phenyl]methylidene}pyridine-3-carbohydrazide (**12**)

Yellow powder, yield: 88%, M.p.: 164 °C; IR: 3423, 3195 (N-H), 3046 (CH, arom.), 2970 (CH, aliph.), 1739 (C=O), 1595 (C=N), 1217, 1106 (C-OC), 1025 (N-N); ^1^H NMR (600 MHz, DMSO-*d_6_*) δ (ppm): 1.96–1.99 (m, 8H, 4xCH_2-pyrrolidine_), 6.60–6.61 (d, 1H, ArH, *J* = 6 Hz), 6.63–6.65 (d, 2H, ArH, *J* = 12 Hz), 7.54–7.56 (d, 1H, ArH, *J* = 12 Hz), 7.67–7.69 (d, 2H, ArH, *J* = 12 Hz), 8.15–8.24 (m, 1H, ArH), 8.29 (s, 1H, =CH), 9.65–9.66 (d, 1H, ArH, *J* = 6 Hz), 11.69 (s, 1H, NH); ^13^C NMR (150 MHz, DMSO-*d_6_*) δ (ppm): 25.38 (2xCH_2-pyrrolidine_), 47.83 (2xCH_2-pyrrolidine_), 111.74, 112.06, 123.94, 124.73, 129.18, 132.17, 135.11 (9C_ar_), 148.53 (=CH), 152.21 (2C_ar_), 164.76 (C=O); Anal. calc. for C_17_H_18_N_4_O (294.35) (%): C 69.37; H 6.16; N 19.03. Found: C 69.25; H 6.31; N 19.50.

*N*-[(5-nitrofuran-2-yl)methylidene]pyridine-3-carbohydrazide (**13**)

Yellow powder, yield: 94%, M.p.: 238 °C; IR: 3252, 3144 (N-H), 3018 (CH, arom.), 2969 (CH, aliph.), 1656 (C=O), 1557 (C=N), 1279, 1093 (C-OC), 1025 (N-N); ^1^H NMR (600 MHz, DMSO-*d_6_*) δ (ppm): 7.32–7.33 (d, 1H, ArH, *J* = 6 Hz), 7.60–7.62 (m, 1H, ArH), 7.82–7.83 (d, 1H, ArH, *J* = 6 Hz), 8.27–8.28 (d, 1H, ArH, *J* = 6 Hz), 8.40 (s, 1H, =CH), 8.80–8.81 (d, 1H, ArH, *J* = 6 Hz), 9.08 (s, 1H, ArH), 12.41 (s, 1H, NH); ^13^C NMR (150 MHz, DMSO-*d_6_*) δ (ppm): 115.05, 116.23, 124.18, 129.11, 136.08 (5C_ar_), 136.62 (=CH), 149.12, 151.92, 152.48, 153.14 (4C_ar_), 162.52 (C=O); Anal. calc. for C_11_H_8_N_4_O_4_ (260.21) (%): C 50.77; H 3.10; N 21.53. Found: C 50.35; H 3.51; N 21.64.

Representative ^1^H NMR and ^13^C NMR spectra of acylhydrazones of nicotinic acid (**2**–**13**) are shown in Appendix A.

#### 4.1.2. Synthesis of 3-Acetyl-2,5-disubstituted-1,3,4-oxadiazolines (**14**–**25**)

The acetic anhydride (10 mL) was added to the acylhydrazones (**2**–**13**) (0.01 mole) obtained earlier. The cyclization reaction was carried out at reflux for 3 h. After that, the acetic anhydride was distilled off under reduced pressure and crushed ice was added to the flask. The content of the flask was shaken variously until crystals precipitated, and left at room temperature for 24 h. The precipitate formed was filtered off under reduced pressure and recrystallized from ethanol. The purity of the obtained compounds was checked by the TLC chromatography.

Detailed physicochemical properties of new derivatives of 3-acetyl-2,5-disubstituted-1,3,4-oxadiazoline (**14–25**):

1-[2-(4-*tert*-butylphenyl)-5-(pyridin-3-yl)-1,3,4-oxadiazol-3(2*H*)-yl]ethan-1-one (**14**)

Creamy powder, yield: 67%, M.p.: 84 °C; IR: 3151 (CH, arom.), 2960 (CH, aliph.), 1675 (C=O), 1593 (C=N), 1283, 1015 (C-OC); ^1^H NMR (600 MHz, DMSO-*d_6_*) δ (ppm): 1.30 (s, 9H, 3xCH_3_), 2.28 (s, 3H, CH_3_), 7.18 (s, 1H, CH_oxadiazole_), 7.41–7.42 (d, 1H, ArH, *J* = 6 Hz), 7.46–7.48 (d, 1H, ArH, *J* = 12 Hz), 7.51–7.53 (d, 2H, ArH, *J* = 12 Hz), 7.67–7.68 (d, 1H, ArH, *J* = 6 Hz), 7.87–7.89 (d, 2H, ArH, *J* = 12 Hz), 8.09–8.10 (d, 1H, ArH, *J* = 6 Hz); ^13^C NMR (150 MHz, DMSO-*d_6_*) δ (ppm): 21.69 (CH_3_), 31.33 (3xCH_3_), 35.24 (C*_t_*_-butyl_), 92.64 (CH_oxadiazole_), 125.82, 126.90, 127.18, 128.49, 129.64, 134.54, 147.70 (9C_ar_), 156.26 (C_oxadiazole_), 164.92, 167.27 (2C_ar_), 167.70 (C=O); Anal. calc. for C_13_H_21_N_3_O_2_ (323.39) (%): C 70.57; H 6.55; N 12.99. Found: C 70.88; H 6.56; N 12.88.

1-[2-(3-bromo-4-hydroxyphenyl)-5-(pyridin-3-yl)-1,3,4-oxadiazol-3(2*H*)-yl]ethan-1-one (**15**)

Brown powder, yield: 75%, M.p.: 68 °C; IR: 3460 (OH), 3015 (CH, arom.), 2970 (CH, aliph.), 1739 (C=O), 1556 (C=N), 1217, 1064 (C-OC); ^1^H NMR (600 MHz, DMSO-*d_6_*) δ (ppm): 2.42 (s, 3H, CH_3_), 7.22 (s, 1H, CH_oxadiazole_), 7.56–7.59 (m, 2H, ArH), 8.01–8.06 (m, 1H, ArH), 8.20–8.22 (m, 1H, ArH), 8.26–8.33 (m, 1H, ArH), 8.77–8.78 (d, 1H, ArH, *J* = 6 Hz), 9.02 (s, 1H, ArH), 7.95 (s, 1H, OH); ^13^C NMR (150 MHz, DMSO-*d_6_*) δ (ppm): 21.73 (CH_3_), 90.69 (CH_oxadiazole_), 112.62, 118.06, 120.89, 124.52, 131.52, 134.12, 134.74, 147.13, 147.86, 152.84, (10C_ar_), 153.37 (C_oxadiazole_), 167.61 (C_ar_), 168.00 (C=O); Anal. calc. for C_15_H_12_BrN_3_O_3_ (362.18) (%): C 49.74; H 3.34; N 11.60. Found: C 49.25; H 3.31; N 11.50.

1-[2-(3,5-dibromo-4-hydroxyphenyl)-5-(pyridin-3-yl)-1,3,4-oxadiazol-3(2*H*)-yl]ethan-1-one (**16**)

Creamy powder, yield: 84%, M.p.: 116 °C; IR: 3456 (OH), 3016 (CH, arom.), 2970 (CH, aliph.), 1739 (C=O), 1570 (C=N), 1217, 1046 (C-OC); ^1^H NMR (600 MHz, DMSO-*d_6_*) δ (ppm): 2.34 (s, 3H, CH_3_), 7.24 (s, 1H, CH_oxadiazole_), 7.37–7.39 (d, 1H, ArH, *J* = 12 Hz), 7.57–7.60 (m, 2H, ArH), 7.90–7.91 (d, 1H, ArH, *J* = 6 Hz), 8.20–8.22 (m, 1H, ArH), 8.77–8.78 (m, 1H, ArH), 9.02 (s, 1H, OH); ^13^C NMR (150 MHz, DMSO-*d_6_*) δ (ppm): 21.71 (CH_3_), 91.30 (CH_oxadiazole_), 116.42, 120.89, 124.56, 125.23, 127.95, 132.05, 134.69, 136.46, 147.79 (9C_ar_), 152.89 (C_oxadiazole_), 153.38, 167.72 (2C_ar_), 168.71 (C=O); Anal. calc. for C_15_H_11_BrN_3_O_3_ (441.07) (%): C 40.85; H 2.51; N 9.53. Found: C 40.27; H 2.31; N 9.80.

1-[2-(2-hydroxy-3,5-diiodophenyl)-5-(pyridin-3-yl)-1,3,4-oxadiazol-3(2*H*)-yl]ethan-1-one (**17**)

Creamy powder, yield: 74%, M.p.: 186 °C; IR: 3456 (OH), 3016 (CH, arom.), 2970 (CH aliph.), 1739 (C=O), 1588 (C=N), 1229, 1067 (C-OC); ^1^H NMR (600 MHz, DMSO-*d_6_*) δ (ppm): 2.20 (s, 3H, CH_3_), 7.16 (s, 1H, CH_oxadiazole_), 7.52–7.61 (m, 2H, ArH), 7.99–8.00 (d, 1H, ArH, *J* = 6 Hz), 8.16–8.17 (d, 1H, ArH, *J* = 6 Hz), 8.34–8.35 (d, 1H, ArH, *J* = 6 Hz), 8.79–8.80 (m, 1H, ArH), 8.97 (s, 1H, OH); ^13^C NMR (150 MHz, DMSO-*d_6_*) δ (ppm): 21.50 (CH_3_), 90.32 (CH_oxadiazole_), 92.94, 97.26, 120.61, 124.70, 134.53, 137.41, 139.36, 147.64, 148.44, 150.67 (10C_ar_), 153.03 (C_oxadiazole_), 153.76 (C_ar_), 167.65 (C=O); Anal. calc. for C_15_H_11_I_2_N_3_O_3_ (535.08) (%): C 33.67; H 2.07; N 7.85. Found: C 32.89; H 2.15; N 7.50.

1-[2-(2-bromo-3-hydroxy-4-methoxyphenyl)-5-(pyridin-3-yl)-1,3,4-oxadiazol-3(2*H*)-yl]ethan-1-one (**18**)

Yellow powder, yield: 64%, M.p.: 124 °C; IR: 3239 (OH), 3018 (CH, arom.), 2887 (CH aliph.), 1667 (C=O), 1592 (C=N), 1286, 1015 (C-OC); ^1^H NMR (600 MHz, DMSO-*d_6_*) δ (ppm): 2.38 (s, 3H, CH_3_), 3.94 (s, 3H, OCH_3_), 7.11 (s, 1H, CH_oxadiazole_), 7.15–7.17 (d, 2H, ArH, *J* = 12 Hz), 7.37–7.38 (d, 1H, ArH, *J* = 6 Hz), 7.41–7.43 (d, 2H, ArH, *J* = 12 Hz), 7.83–7.85 (d, 1H, ArH, *J* = 12 Hz), 10.12 (s, 1H, OH); ^13^C NMR (150 MHz, DMSO-*d_6_*) δ (ppm): 21.89 (CH_3_), 56.94 (OCH_3_), 100.52 (CH_oxadiazole_), 110.92, 113.84, 120.60, 122.47, 127.13, 128.28, 135.71, 144.52, 149.04, 151.34 (10C_ar_), 153.83 (C_oxadiazole_), 156.78 (C_ar_), 165.14 (C=O); Anal. calc. for C_16_H_14_BrN_3_O_4_ (392.20) (%): C 49.00; H 3.60; N 10.71. Found: C 49.25; H 3.20; N 10.50.

1-[2-(2-bromo-5-fluorophenyl)-5-(pyridin-3-yl)-1,3,4-oxadiazol-3(2*H*)-yl]ethan-1-one (**19**)

White powder, yield: 74%, M.p.: 118 °C; IR: 3016 (CH, arom.), 2970 (CH aliph.), 1739 (C=O), 1588 (C=N), 1217, 1031 (C-OC); ^1^H NMR (600 MHz, DMSO-*d_6_*) δ (ppm): 2.32 (s, 1H, CH_3_), 7.29 (s, 1H, CH_oxadiazole_), 7.32–7.35 (m, 1H, ArH), 7.39–7.41 (m, 1H, ArH), 7.56–7.58 (m, 1H, ArH), 7.79–7.82 (m, 1H, ArH), 8.16–8.18 (m, 1H, ArH), 8.76–8.77 (d, 1H, ArH, *J* = 6 Hz), 8.98 (s, 1H, ArH); ^13^C NMR (150 MHz, DMSO-*d_6_*) δ (ppm): 21.66 (CH_3_), 92.21 (CH_oxadiazole_), 117.22, 119.79, 120.87, 124.58, 134.62, 135.85, 136.79, 147.70 (8C_ar_), 152.87 (C_oxadiazole_), 153.07, 161.25, 162.88 (3C_ar_), 167.82 (C=O); Anal. calc. for C_15_H_11_BrFN_4_O_2_ (364.17) (%): C 49.47; H 3.04; N 11.52. Found: C 49.88; H 3.08; N 11.99.

1-[2-(4-bromo-2-fluorophenyl)-5-(pyridin-3-yl)-1,3,4-oxadiazol-3(2*H*)-yl]ethan-1-one (**20**)

White powder, yield: 68%, M.p.: 100 °C; IR: 3016 (CH, arom.), 2970 (CH aliph.), 1683 (C=O), 1592 (C=N), 1212, 1042 (C-OC); ^1^H NMR (600 MHz, DMSO-*d_6_*) δ (ppm): 2.27 (s, 3H, CH_3_), 7.32 (s, 1H, CH_oxadiazole_), 7.53–7.56 (m, 2H, ArH), 7.57–7.59 (m, 2H, ArH), 7.68–7.70 (d, 1H, ArH, *J* = 12 Hz), 8.18–8.20 (m, 1H, ArH), 8.77–8.78 (m, 1H, ArH), 8.99 (s, 1H, ArH); ^13^C NMR (150 MHz, DMSO-*d_6_*) δ (ppm): 21.56 (CH_3_), 88.72 (CH_oxadiazole_), 120.05, 120.82, 123.10, 124.61, 128.56, 131.67, 134.57, 147.65 (8C_ar_), 152.92 (C_oxadiazole_), 153.28, 159.78, 161.46 (3C_ar_), 167.34 (C=O); Anal. calc. for C_15_H_11_BrFN_3_O_2_ (364.17) (%): C 49.47; H 3.04; N 11.52. Found: C 49.25; H 3.02; N 11.50.

1-[2-(2-chloro-6-fluorophenyl)-5-(pyridin-3-yl)-1,3,4-oxadiazol-3(2*H*)-yl]ethan-1-one (**21**)

Creamy powder, yield: 71%, M.p.: 102 °C; IR: 3049 (CH, arom.), 2970 (CH aliph.), 1655 (C=O), 1577 (C=N), 1210, 1030 (C-OC); ^1^H NMR (600 MHz, DMSO-*d_6_*) δ (ppm): 2.23 (s, 3H, CH_3_), 7.32–7.35 (m, 1H, ArH), 7.43–7.44 (d, 1H, ArH, *J* = 6 Hz), 7.54–7.58 (m, 2H, ArH), 7.57 (s, 1H, CH_oxadiazole_), 8.20–8.22 (m, 1H, ArH), 8.77–8.78 (m, 1H, ArH), 9.01 (s, 1H, ArH); ^13^C NMR (150 MHz, DMSO-*d_6_*) δ (ppm): 21.44 (CH_3_), 87.26 (CH_oxadiazole_), 116.03, 120.83, 121.25, 124.62, 126.95, 133.39, 134.52, 147.61 (8C_ar_), 152.88 (C_oxadiazole_), 153.10, 161.23, 162.91 (3C_ar_), 167.09 (C=O); Anal. calc. for C_15_H_11_ClFN_3_O_2_ (319.72) (%): C 56.35; H 3.47; N 13.14. Found: C 55.99; H 3.62; N 13.49.

1-[2-(2-chloro-5-nitrophenyl)-5-(pyridin-3-yl)-1,3,4-oxadiazol-3(2*H*)-yl]ethan-1-one (**22**)

Grey powder, yield: 88%, M.p.: 168 °C; IR: 3016 (CH, arom.), 2970 (CH aliph.), 1676 (C=O), 1518 (C=N), 1215, 1020 (C-OC); ^1^H NMR (600 MHz, DMSO-*d_6_*) δ (ppm): 2.32 (s, 3H, CH_3_), 7.49 (s, 1H, CH_oxadiazole_), 7.57–7.59 (m, 1H, ArH), 7.90–7.92 (d, 1H, ArH, *J* = 12 Hz), 8.19–8.21 (m, 1H, ArH), 8.32–8.36 (m, 2H, ArH), 8.78–8.79 (d, 1H, ArH, *J* = 6 Hz), 9.00 (s, 1H, ArH); ^13^C NMR (150 MHz, DMSO-*d_6_*) δ (ppm): 21.57 (CH_3_), 90.54 (CH_oxadiazole_), 120.69, 124.62, 125.31, 126.97, 132.55, 134.68, 139.42, 147.16, 147.74, 150.26 (10C_ar_), 153.00 (C_oxadiazole_), 153.35 (C_ar_), 167.98 (C=O); Anal. calc. for C_15_H_11_ClN_4_O_4_ (346.72) (%): C 51.96; H 3.20; N 16.16. Found: C 49.85; H 3.21; N 17.03.

1-[2-(2,3-difluorophenyl)-5-(pyridin-3-yl)-1,3,4-oxadiazol-3(2*H*)-yl]ethan-1-one (**23**)

Brown powder, yield: 63%, M.p.: 88 °C; IR: 3035 (CH, arom.), 2970 (CH aliph.), 1666 (C=O), 1591 (C=N), 1210, 1019 (C-OC); ^1^H NMR (600 MHz, DMSO-*d_6_*) δ (ppm): 2.28 (s, 3H, CH_3_), 7.30–7.33 (m, 1H, ArH), 7.38 (s, 1H, CH_oxadiazole_), 7.40–7.41 (m, 1H, ArH), 7.57–7.59 (m, 2H, ArH), 8.20–8.21 (m, 1H, ArH), 8.78–8.79 (m, 1H, ArH), 9.01 (s, 1H, ArH); ^13^C NMR (150 MHz, DMSO-*d_6_*) δ (ppm): 21.57 (CH_3_), 88.52 (CH_oxadiazole_), 119.82, 119.93, 120.77, 124.62, 125.12, 125.85, 126.01, 126.06, 134.61, 147.68 (10C_ar_), 152.95 (C_oxadiazole_), 153.28 (C_ar_), 167.42 (C=O); Anal. calc. for C_15_H_11_F_2_N_3_O_2_ (303.26) (%): C 59.41; H 3.66; N 13.86. Found: C 59.25; H 3.31; N 13.42.

1-[5-(pyridin-3-yl)-2-[4-(pyrrolidin-1-yl)phenyl]-1,3,4-oxadiazol-3(2*H*)-yl]ethan-1-one (**24**)

Brown powder, yield: 72%, M.p.: 62 °C; IR: 3025 (CH, arom.), 2970 (CH aliph.), 1652 (C=O), 1593 (C = N), 1217, 1118 (C-OC); ^1^H NMR (600 MHz, DMSO-*d_6_*) δ (ppm): 2.13 (s, 3H, CH_3_), 2.39–2.40 (m, 4H, 2xCH_2-pyrrolidine_), 2.61–2.63 (m, 4H, 2xCH_2-pyrrolidine_), 7.30–7.33 (m, 2H, ArH), 7.38 (s, 1H, CH_oxadiazole_), 7.40–7.41 (m, 1H, ArH), 7.55–7.59 (m, 2H, ArH), 8.20–8.21 (m, 1H, ArH), 8.78–8.79 (m, 1H, ArH), 9.01 (s, 1H, ArH); ^13^C NMR (150 MHz, DMSO-*d_6_*) δ (ppm): 21.52 (CH_3_), 25.38 (2xCH_2-pyrrolidine_), 47.84 (2xCH_2-pyrrolidine_), 101.3 (CH_oxadiazole_), 107.20, 111.75, 124.09, 124.72, 128.28, 132.16, 135.71, 148.25 (10C_ar_), 152.15 (C_oxadiazole_), 158.12 (C_ar_), 166.20 (C=O); Anal. calc. for C_19_H_20_N_4_O_2_ (336.39) (%): C 67.84; H 5.99; N 16.66. Found: C 68.02; H 6.14; N 16.54.

1-[2-(5-nitrofuran-2-yl)-5-(pyridin-3-yl)-1,3,4-oxadiazol-3(2*H*)-yl]ethan-1-one (**25**)

Brown powder, yield: 87%, M.p.: 108 °C; IR: 3016 (CH, arom.), 2970 (CH aliph.), 1739 (C=O), 1541 (C=N), 1217, 1016 (C-OC); ^1^H NMR (600 MHz, DMSO-*d_6_*) δ (ppm): 2.31 (s, 3H, CH_3_), 7.27–7.28 (d, 1H, ArH, *J* = 6 Hz), 7.41 (s, 1H, CH_oxadiazole_), 7.59–7.61 (m, 1H, ArH), 7.73–7.74 (d, 1H, ArH, *J* = 6 Hz), 8.21–8.23 (m, 1H, ArH), 8.79–8.80 (m, 1H, ArH), 9.01–9.02 (d, 1H, ArH, *J* = 6 Hz); ^13^C NMR (150 MHz, DMSO-*d_6_*) δ (ppm): 21.60 (CH_3_), 85.14 (CH_oxadiazole_), 113.67, 115.18, 120.48, 124.67, 134.70, 147.74, 150.74, 152.28 (8C_ar_), 153.14 (C_oxadiazole_), 154.31 (C_ar_), 167.84 (C=O); Anal. calc. for C_13_H_10_N_4_O_5_ (302.24) (%): C 51.66; H 3.33; N 18.54. Found: C 49.25; H 3.51; N 18.50.

Representative ^1^H NMR and ^13^C NMR spectra of 3-acetyl-2,5-disubstituted-1,3,4-oxadiazoline (**14**–**25**) are shown in Appendix A.

### 4.2. Microbiology

All of the obtained compounds were tested for their antimicrobial activity by the broth microdilution method according to the procedures described earlier by our research group [58,59,60,61,62]. All tests were in the line with the European Committee on Antimicrobial Susceptibility Testing (EUCAST) [63] and Clinical and Laboratory Standards Institute [64] recommendations. In the assays, we used a panel of reference and clinical or saprophytic strains of microorganisms from the American Type Culture Collection (ATCC) in microbiology assays. All of the experiments were repeated three times, and representative data are presented. Stock solutions were prepared by dissolving synthesized substances in DMSO.

### 4.3. Cytotoxicity Studies

#### 4.3.1. Cell Cultures

Human colon adenocarcinoma cell line HT29 (ATCC No. HTB-38), derived from a grade I tumour, was cultured in RPMI 1640 medium supplemented with 10% fetal calf serum (FCS) (GibcoTM, Paisley, UK) and antibiotics (100 U/mL penicillin and 100 µg/mL streptomycin) (Sigma, St. Louis, MO, USA) at 37 °C in a humidified atmosphere with 5% CO_2_. Human normal colon epithelial cells CCD 841 CoTr (ATCC No. CRL-1807) were cultured in RPMI 1640 + DMEM (1:1) medium (Sigma, St. Louis, MO, USA) supplemented with 10% FCS and antibiotics at 34 °C in a 5% CO_2_/95% air atmosphere.

#### 4.3.2. MTT Assay

After 24 h incubation of cells with the tested compounds in 100 µL of culture medium, MTT solution (5 mg/mL, 25 µL/well) was added and incubated for an additional 3 h. The purple crystals of formazan which formed in the medium were solubilized overnight in 10% sodium dodecyl sulphate (SDS) in a 0.01 M HCl mixture. The product was quantified spectrophotometrically by measuring its absorbance at 570 nm with the use of an Emax Microplate Reader (Molecular Devices Corporation, Menlo Park, CA).

#### 4.3.3. Neutral Red (NR) Uptake Assay

Cells were grown for 24 h in a 96-well multiplate in 100 µL of culture medium (RPMI 1640) supplemented with 5% fetal bovine serum (FBS) and tested compounds. Subsequently, the medium was discarded and a 0.4% neutral red solution in 2% FBS medium was added to each well. The plate was incubated for 3 h at 37 °C in a humidified 5% CO_2_/95% air incubator. After incubation, the dye-containing medium was removed, the cells were fixed with 1% CaCl_2_ in 4% paraformaldehyde, and the incorporated dye was solubilized using 1% acetic acetate in a 50% ethanol solution (100 µL). The plates were gently shaken for 20 min at room temperature and the absorbance of the extracted dye was measured spectrophotometrically at 540 nm.

#### 4.3.4. Nitric Oxide (NO) Measurement

Nitrate, i.e., a stable end product of NO, was determined in culture supernatants with a spectrophotometric method based on the Griess reaction. Culture supernatants were collected from cell cultures treated with specific concentrations of the tested compounds. Briefly, 100 µL of the culture supernatant was placed in 96-well flat-bottomed plates in triplicate and incubated with 100 µL of Griess reagent (1% sulfanilamide/0.1% *N*-(1-naphthyl)ethylenediamine dihydrochloride) in 3% H_3_PO_4_ at room temperature for 10 min. The optical density was measured at 550 nm using a microplate reader. A standard curve was prepared with the use of 0.5–25 µM sodium nitrite (NaNO_2_) for calibration.

#### 4.3.5. DPPH Free Radical Scavenging Test

Free radical scavenging activity of terpene was measured by the DPPH assay. Briefly, 100 µL of DPPH solution (0.2 mg/mL in ethanol) was added to 100 µL of the tested compound concentrations (0–200 µg/mL). Trolox in increasing concentrations (1–50 µg/mL) was used as a standard. After 20 min of incubation at room temperature, the absorbances of the solutions were measured at 515 nm. The lower the absorbance, the higher the free radical scavenging activity of the compounds. The activity of each compound was determined by comparing its absorbance with that of the standard.

The ability of the compounds to scavenge the DPPH radical was calculated by the following formula:DPPH scavenging effect (%) = [(Xcontrol − Xcompound/Xcontrol) × 100] 

Xcontrol is the absorbance of the control and Xcompound is the absorbance in the presence of synthesized compounds.

#### 4.3.6. Ferric-Reducing Antioxidant Power Assay

Each compound concentration was dissolved in Milli-Q water and mixed with an equal volume of 0.2 M sodium phosphate buffer (pH 6.6) and 1% potassium ferricyanide. The mixture was incubated for 30 min at 37 °C. Thereafter, 10% trichloroacetic acid (*w/v*) was added and the mixture was centrifuged at 1000× *g* for 5 min. One millilitre of the upper layer was mixed with an equal volume of Milli-Q water and 0.1% ferric chloride. The absorbance was read at 700 nm with the use of an EL800 Universal Microplate Reader (BioTek Instruments, Winooski, VT, USA). Ascorbic acid (0–150 µg/mL) was used as a positive control.

#### 4.3.7. May–Grünwald–Giemsa (MGG) Staining

The cells were incubated in 24-well plates in 1 mL of culture medium supplemented with tested compounds. After 24 h of incubation (37 °C in a humidified 5% CO_2_/95% air), the medium was discarded and the cell cultures were rinsed with RPMI 1640 medium and stained with May–Grünwald (MG) stain for 5 min followed by staining for another 5 min in MG diluted in an equal quantity of water. The MG was removed and Giemsa reagent (diluted 1:20 in water) was added to the cells, which were next incubated at room temperature for 15 min. After that, the cells were rinsed three times with water, dried and subjected to microscopic observations (Olympus, BX51; Olympus).

#### 4.3.8. Statistical Analysis

Results are presented as mean ± SD from three experiments. Data were analysed with the use of one-way ANOVA with Dunnett’s post hoc test. Differences of *p* ≤ 0.05 were considered as significant.

### 4.4. Molecular Docking

A molecular docking study was carried out using the Autodock Vina 4.2.

The X-ray crystal structures of target proteins were downloaded from the Protein Data Bank. The AutoDock tools were used to remove water molecules, add polar hydrogen atoms, merge nonpolar hydrogen atoms, define rotatable bonds, and add Kollman charges. The validations of selected docking parameters were performed by redocking of the initial ligands from the used enzyme structures.

The structure of nitrofurantoin was downloaded from the PubChem portal. The 3D structures of the synthesized compounds were prepared with the use of HyperChem 7.5 software. At first, the molecules were optimized by the method of molecular mechanics MM + with the achievement of an RMS gradient of less than 0.1 kcal/(mol Å). The final minimization of the energies of the investigated intermediates was carried out by the semi-empirical quantum chemical method PM3 until the RMS gradient was less than 0.01 kcal/(mol Å). The visualization and interpretation of the obtained data were performed with the use of Discovery Studio Visualizer.

Theoretical predictions of AMDET profiles were performed using SwissADME (http://www.swissadme.ch/) (accessed on 21 February 2022) and ProTOX (https://tox-new.charite.de/protox_II/) portals (accessed on 21 February 2022).

## 5. Conclusions

In summary, in this paper, we presented the synthesis, results of antimicrobial activity and cytotoxicity assays and molecular docking study of twenty-four new nicotinic acid derivatives. The presented synthesis was carried out in two stages. In the first stage, as a result of the condensation reaction, we obtained twelve acylhydrazones (**2**–**13**), which were subjected to cyclization reaction in acetic anhydride to obtain twelve new 3-acetyl-2,5-disubstituted-1,3,4-oxadiazoline derivatives (**14**–**25**). All of the obtained substances were subjected to a series of in vitro antimicrobial activity assays which showed that twelve of the synthesized compounds had antimicrobial activity against the tested strains. Importantly, four of the obtained compounds showed activity towards the resistant MRSA strain (*Staphylococcus aureus* ATCC 43300) within the range of MIC = 7.81–31.25 µg/mL. Additionally, on the basis of the presented data, it can be concluded that acylhydrazones showed better antibacterial activity in relation to the corresponding 3-acetyl-1,3,4-oxadiazoline derivatives. The most active of the tested compounds, i.e., compound **13** with the 5-nitrofuran substituent, showed the highest activity against *Staphylococcus epidermidis* ATCC 12228 (MIC = 1.95 µg/mL), *S. aureus* ATCC 6538 (MIC = 3.91 µg/mL) as well as high activity against the MRSA *S. aureus* ATCC 43300 strain (MIC = 7.81 µg/mL). Selected substances have also undergone cytotoxicity tests which showed that they are safe for normal cell lines. Additionally, molecular docking tests confirmed the obtained results.

## Data Availability

Data are contained within the article.

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
