# Peer review of "Synthesis, Biological Activity and Molecular Docking Studies of Novel Nicotinic Acid Derivatives"

_ijms, 2022, doi:10.3390/ijms23052823_

Round 1

Reviewer 1 Report

The manuscript by Paruch is very interesting because it deals with a relevant issue that is the AMR.

The manuscript is well written even if some typos need to be fixed.

However, some concerns arose that need to be addressed before acceptance. They are listed below:

1)in the introduction the list of resistant microorganisms would benefit from doing a table that includes the name/antibiotic/type of caused diseases/refs.

2)results/discussion: The microbiology results should be better discussed in terms of the difference between gram positive and gram negative. Moreover, why do the authors include also fungi? this should be anticipated also in the introduction. the SAR analysis of the most effective compounds should be better presented. 

3)results/discussion: the rationale of cytotoxicity studies should be better explained in the introduction and discussed in the discussion section. why do the authors decide on those cell lines? 

4)results/discussion: the choice for molecular docking analysis should be better clarified. why did the authors specifically decide for nitroreductase and not other enzymes which are different between gram positive and negative as an example? furthermore, did the authors plan to experimentally test the compounds on the selected enzyme? this should be discussed. moreover, the way of entry of the compounds in cells should be also hypothesized and discussed. 

Author Response

Detailed responses to reviewer’s comments:

Please consider our responses to the First Reviewer:

We would like to thank the Reviewer for comments, which we feel gave us the opportunity to strengthen the message of the manuscript. Please find our responses below.

Reviewer’s comments

General comment:

“The manuscript by Paruch is very interesting because it deals with a relevant issue that is the AMR.

The manuscript is well written even if some typos need to be fixed.

However, some concerns arose that need to be addressed before acceptance. They are listed below:”

Our response: We would like to thank the Reviewer for comments regarding our manuscript and for the possibility to correct our article.

First comment:

“ 1) in the introduction the list of resistant microorganisms would benefit from doing a table that includes the name/antibiotic/type of caused diseases/refs.”

Our response:

Thank you for this comment. We have modified the introduction section of our revised manuscript according to the reviewers’ comment and added some new information and a table about resistant microorganisms.

Second comment:

“2) results/discussion: The microbiology results should be better discussed in terms of the difference between gram positive and gram negative. Moreover, why do the authors include also fungi? this should be anticipated also in the introduction. the SAR analysis of the most effective compounds should be better presented.”

Our response:

Thank you for this suggestion. We have modified the introduction and discussion section of our manuscript and added some new information concerning fungi and SAR analysis.

Third comment:

„3) results/discussion: the rationale of cytotoxicity studies should be better explained in the introduction and discussed in the discussion section. why do the authors decide on those cell lines?”

Our response:

Thank you for this suggestion. We have modified the introduction and discussion section on the basis and in the context of recent literature. Cell lines were chosen on the basis of literature reports.

Fourth comment:

„4) results/discussion: the choice for molecular docking analysis should be better clarified. why did the authors specifically decide for nitroreductase and not other enzymes which are different between gram positive and negative as an example? furthermore, did the authors plan to experimentally test the compounds on the selected enzyme? this should be discussed. moreover, the way of entry of the compounds in cells should be also hypothesized and discussed.”

Our response:

Thank you for this comment. The choice of Nitroreductase for docking simulation is quite clearly described. The presence of antibacterial activity for compounds 5 and 17 with the 5-nitrofuran moiety and the lack of such activity for structurally related 10 and 22 allow to suggest that the activity for both Gram-negative and Gram-positive bacteria’s is connected with the Nitroreductase and 5-nitrofuran core.

Dihydrofolate reductase and tyrosyl-tRNA synthetase were selected for docking research due to literally reported activity of some nicotinic acid and 1,3,4-oxadiazole derivatives for the above-mentioned enzymes [1-4].

References:

  1. Nerkar, A.G., et al., In silico screening, synthesis and in vitro evaluation of some quinazolinone and pyridine derivatives as dihydrofolate reductase inhibitors for anticancer activity. Journal of chemistry 6 (2009): S97-S102, Article ID 506576, https://doi.org/10.1155/2009/506576.
  2. Argyrides A., et al., Mycobacterium tuberculosis dihydrofolate reductase is a target for isoniazid. Nature Structural & Molecular Biology 13(5) (2006): 408-413, doi: 10.1038/nsmb1089
  3. Sun J., Lv P.-C., Zhu H.-L., Tyrosyl-tRNA synthetase inhibitors: a patent review. Expert Opinion on Therapeutic Patents 27(5) (2017): 557-564, doi: 10.1080/13543776.2017.1273350
  4. Thakkar S.S., et al., 1,2,4-Triazole and 1,3,4-oxadiazole analogues: Synthesis, MO studies, in silico molecular docking studies, antimalarial as DHFR inhibitor and antimicrobial activities. Bioorganic & Medicinal Chemistry 25(15) (2017): 4064-4075.

We have also added ADMET profiles to the revised version of our manuscritp.

We hope these responses satisfy the Reviewer’s comment.

We would like to thank the Reviewer for help to make our contribution clearly focused and readable to the audience of the Journal.

Reviewer 2 Report

Dear, Kinga Paruch, Ph.D.

Please check and revise.

-------------------------------------------------------------------------------------------------------------------------

New compounds 2-13

These cpds NMR data were shown in supplementary, but they should be shown in  4.1.1. Synthesis of acylhydrazones of nicotinic acid hydrazide (2-14) section. Instead of them, NMR spectra of them should be shown in supplementary.

-------------------------------------------------------------------------------------------------------------------------

Author Response

Detailed responses to reviewer’s comments:

Please consider our responses to the Second Reviewer:

We would like to thank the Reviewer for comments, which we feel gave us the opportunity to strengthen the message of the manuscript. Please find our responses below.

Reviewer’s comments

General comment:

„New compounds 2-13

These cpds NMR data were shown in supplementary, but they should be shown in  4.1.1. Synthesis of acylhydrazones of nicotinic acid hydrazide (2-14) section. Instead of them, NMR spectra of them should be shown in supplementary.”

Our response:

Thank you very much for comments regarding our manuscript and for the possibility to correct our article. We have modified the text of the revised version of our manuscript and added representative spectra of obtained compounds to the Supplementary Materials (Figures 1S-8S).

We hope these responses satisfy the Reviewer’s comment.

We would like to thank the Reviewer for help to make our contribution clearly focused and readable to the audience of the Journal.

Round 2

Reviewer 1 Report

authors addressed all my concerns and the manuscript can now be accepted